# Online Adversarial Purification based on Self-Supervised Learning

**Changhao Shi[1], Chester Holtz[2] & Gal Mishne[1,2,3]**
[1]Department of Electrical and Computer Engineering,
[2]Department of Computer Science and Engineering,
[3]The Halıcıoğlu Data Science Institute
University of California, San Diego
`{cshi,chholtz,gmishne}@ucsd.edu`

## Abstract

Deep neural networks are known to be vulnerable to adversarial examples, where a perturbation in the input space leads to an amplified shift in the latent network representation. In this paper, we combine canonical supervised learning with self-supervised representation learning, and present Self-supervised Online Adversarial Purification (SOAP), a novel defense strategy that uses a self-supervised loss to purify adversarial examples at test-time. Our approach leverages the label-independent nature of self-supervised signals, and counters the adversarial perturbation with respect to the self-supervised tasks. SOAP yields competitive robust accuracy against state-of-the-art adversarial training and purification methods, with considerably less training complexity. In addition, our approach is robust even when adversaries are given knowledge of the purification defense strategy. To the best of our knowledge, our paper is the first that generalizes the idea of using self-supervised signals to perform online test-time purification.

## 1 Introduction

Deep neural networks have achieved remarkable results in many machine learning applications. However, these networks are known to be vulnerable to adversarial attacks, i.e. strategies which aim to find adversarial examples that are close or even perceptually indistinguishable from their natural counterparts but easily mis-classified by the networks. This vulnerability raises theory-wise issues about the interpretability of deep learning as well as application-wise issues when deploying neural networks in security-sensitive applications.

Many strategies have been proposed to empower neural networks to defend against these adversaries. The current most widely used genre of defense strategies is adversarial training. Adversarial training is an on-the-fly data augmentation method that improves robustness by training the network not only with clean examples but adversarial ones as well. For example, Madry et al. (2017) propose projected gradient descent as a universal first-order attack and strengthen the network by presenting it with such adversarial examples during training (e.g., adversarial training). However, this method is computationally expensive as finding these adversarial examples involves sample-wise gradient computation at every epoch.

Self-supervised representation learning aims to learn meaningful representations of unlabeled data where the supervision comes from the data itself. While this seems orthogonal to the study of adversarial vulnerability, recent works use representation learning as a lens to understand as well as improve adversarial robustness (Hendrycks et al., 2019; Mao et al., 2019; Chen et al., 2020a; Naseer et al., 2020). This recent line of research suggests that self-supervised learning, which often leads to a more informative and meaningful data representation, can benefit the robustness of deep networks.

In this paper, we study how self-supervised representation learning can improve adversarial robustness. We present Self-supervised Online Adversarial Purification (SOAP), a novel defense strategy that uses an auxiliary self-supervised loss to purify adversarial examples at test-time, as illustrated in Figure 1. During training, beside the classification task, we jointly train the network on a carefully selected self-supervised task. The multi-task learning improves the robustness of the network

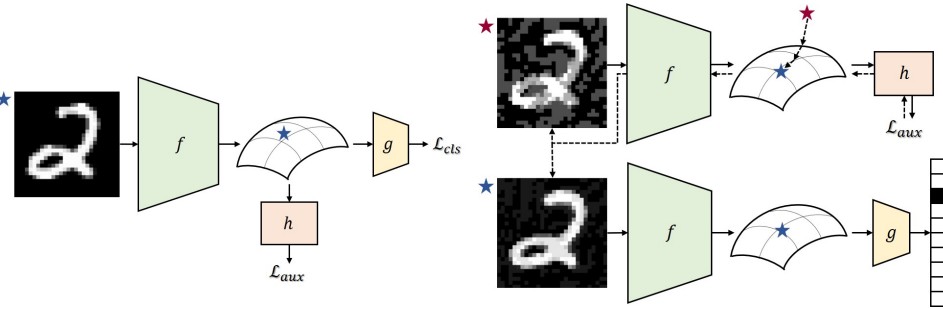

(a) Joint training of classification and auxiliary.     (b) Test-time online purification

Figure 1: An illustration of self-supervised online adversarial purification (SOAP). Left: joint training of the classification and the auxiliary task; Right: input adversarial example is purified iteratively to counter the representational shift, then classified. Note that the encoder is shared by both classification and purification.

and more importantly, enables us to counter the adversarial perturbation at test-time by leveraging the label-independent nature of self-supervised signals. Experiments demonstrate that SOAP performs competitively on various architectures across different datasets with only a small computation overhead compared with vanilla training. Furthermore, we design a new attack strategy that targets both the classification and the auxiliary tasks, and show that our method is robust to this adaptive adversary as well. Code is available at https://github.com/Mishne-Lab/SOAP.

## 2 RELATED WORK

**Adversarial training**   Adversarial training aims to improve robustness through data augmentation, where the network is trained on adversarially perturbed examples instead of the clean original training samples (Goodfellow et al., 2014; Kurakin et al., 2016; Tramèr et al., 2017; Madry et al., 2017; Kannan et al., 2018; Zhang et al., 2019). By solving a min-max problem, the network learns a smoother data manifold and decision boundary which improve robustness. However, the computational cost of adversarial training is high because strong adversarial examples are typically found in an iterative manner with heavy gradient calculation. Compared with adversarial training, our method avoids solving the complex inner-max problem and thus is significantly more efficient in training. Our method does increase test-time computation but it is practically negligible per sample.

**Adversarial purification**   Another genre of robust learning focuses on shifting the adversarial examples back to the clean data representation , namely purification. Gu & Rigazio (2014) exploited using a general DAE (Vincent et al., 2008) to remove adversarial noises; Meng & Chen (2017) train a reformer network, which is a collection of autoencoders, to move adversarial examples towards clean manifold; Liao et al. (2018) train a UNet that can denoise adversarial examples to their clean counterparts; Samangouei et al. (2018) train a GAN on clean examples and project the adversarial examples to the manifold of the generator; Song et al. (2018) assume adversarial examples have lower probability and learn the image distribution with a PixelCNN so that they can maximize the probability of a given test example; Naseer et al. (2020) train a conditional GAN by letting it play a min-max game with a critic network in order to differentiate between clean and adversarial examples. In contrast to above approaches, SOAP achieves better robust accuracy and does not require a GAN which is hard and inefficient to train. More importantly, our approach exploits a wider range of self-supervised signals for purification and conceptually can be applied to any format of data and not just images, given an appropriate self-supervised task.

**Self-supervised learning**   Self-supervised learning aims to learn intermediate representations of unlabeled data that are useful for unknown downstream tasks. This is done by solving a self-supervised task, or pretext task, where the supervision of the task comes from the data itself. Recently, a variety of self-supervised tasks have been proposed on images, including data reconstruction (Vincent et al., 2008; Rifai et al., 2011), relative positioning of patches (Doersch et al., 2015; Noroozi & Favaro, 2016), colorization (Zhang et al., 2016), transformation prediction (Dosovitskiy et al., 2014; Gidaris et al., 2018) or a combination of tasks (Doersch & Zisserman, 2017).

More recently, studies have shown how self-supervised learning can improve adversarial robustness. Mao et al. (2019) find that adversarial attacks fool the networks by shifting latent representation to

a false class. Hendrycks et al. (2019) observe that PGD adversarial training along with an auxiliary rotation prediction task improves robustness, while Naseer et al. (2020) use feature distortion as a self-supervised signal to find transferable attacks that generalize across different architectures and tasks. Chen et al. (2020a) combine adversarial training and self-supervised pre-training to boost fine-tuned robustness. These methods typically combine self-supervised learning with adversarial training, thus the computational cost is still high. In contrast, our approach achieves robust accuracy by test-time purification which uses a variety of self-supervised signals as auxiliary objectives.

## 3 SELF-SUPERVISED PURIFICATION

### 3.1 PROBLEM FORMULATION

As aforementioned, Mao et al. (2019) observe that adversaries shift clean representations towards false classes to diminish robust accuracy. The small error in input space, carefully chosen by adversaries, gets amplified through the network, and finally leads to wrong classification. A natural way to solve this is to perturb adversarial examples so as to shift their representation back to the true classes, i.e. purification. In this paper we only consider classification as our main task, but our approach should be easily generalized to other tasks as well.

Consider an encoder $z = f(x; \theta_{\text{enc}})$, a classifier $g(z; \theta_{\text{cls}})$ on top of the representation $z$, and the network $g \circ f$ a composition of the encoder and the classifier. We formulate the purification problem as follows: for an adversarial example $(x_{\text{adv}}, y)$ and its clean counterpart $(x, y)$ (unknown to the network), a purification strategy $\pi$ aims to find $x_{\text{pfy}} = \pi(x_{\text{adv}})$ that is as close to the clean example $x$ as possible: $x_{\text{pfy}} \to x$. However, this problem is underdetermined as different clean examples can share the same adversarial counterpart, i.e. there might be multiple or even infinite solutions for $x_{\text{pfy}}$. Thus, we consider the relaxation

$$\min_{\pi} \mathcal{L}_{\text{cls}}\left((g \circ f)(x_{\text{pfy}}), y\right) \quad \text{s.t. } ||x_{\text{pfy}} - x_{\text{adv}}|| \le \epsilon_{\text{adv}}, \quad x_{\text{pfy}} = \pi(x_{\text{adv}}), \tag{1}$$

i.e. we accept $x_{\text{pfy}}$ as long as $\mathcal{L}_{\text{cls}}$ is sufficiently small and the perturbation is bounded. Here $\mathcal{L}_{\text{cls}}$ is the cross entropy loss for classification and $\epsilon_{\text{adv}}$ is the budget of adversarial perturbation. However, this problem is still unsolvable since neither the true label $y$ nor the budget $\epsilon_{\text{adv}}$ is available at test-time. We need an alternative approach that can lead to a similar optimum.

### 3.2 SELF-SUPERVISED ONLINE PURIFICATION

Let $h(z; \theta_{\text{aux}})$ be an auxiliary device that shares the same representation $z$ with $g(z; \theta_{\text{cls}})$, and $\mathcal{L}_{\text{aux}}$ be the auxiliary self-supervised objective. The intuition behind SOAP is that the shift in representation $z$ that hinders classification will hinder the auxiliary self-supervised task as well. In other words, large $\mathcal{L}_{\text{aux}}$ often implies large $\mathcal{L}_{\text{cls}}$. Therefore, we propose to use $\mathcal{L}_{\text{aux}}$ as an alternative to $\mathcal{L}_{\text{cls}}$ in Eq. (1). Then we can purify the adversarial examples using the auxiliary self-supervised signals, since the purified examples which perform better on the auxiliary task (small $\mathcal{L}_{\text{aux}}$) should perform better on classification as well (small $\mathcal{L}_{\text{cls}}$).

During training, we jointly minimize the classification loss and self-supervised auxiliary loss

$$\min_{\theta} \{\mathcal{L}_{\text{cls}}\left((g \circ f)(x; \theta_{\text{enc}}, \theta_{\text{cls}}), y\right) + \alpha \mathcal{L}_{\text{aux}}\left((h \circ f)(x; \theta_{\text{enc}}, \theta_{\text{aux}})\right)\}, \tag{2}$$

where $\alpha$ is a trade-off parameter between the two tasks. At test-time, given fixed network parameters $\theta$, we use the label-independent auxiliary objective to perform gradient descent *in the input space*. The purification objective is

$$\min_{\pi} \mathcal{L}_{\text{aux}}((h \circ f)(x_{\text{pfy}})) \quad \text{s.t. } ||x_{\text{pfy}} - x_{\text{adv}}|| \le \epsilon_{\text{pfy}}, x_{\text{pfy}} = \pi(x_{\text{adv}}), \tag{3}$$

where $\epsilon_{\text{pfy}}$ is the budget of purification. This is legitimate at test-time because unlike Eq. (1), the supervision or the purification signal comes from data itself. Also, compared with vanilla training the only training increment of SOAP is an additional self-supervised regularization term. Thus, the computational complexity is largely reduced compared with adversarial training methods. In Sec. 4, we will show that adversarial examples do perform worse on auxiliary tasks and the gradient of the auxiliary loss provides useful information on improving robustness. Note that $\epsilon_{\text{adv}}$ is replaced with $\epsilon_{\text{pfy}}$ in Eq. (3), and we will discuss how to find appropriate $\epsilon_{\text{pfy}}$ in the next section.

| **Algorithm 1** PGD attack | **Algorithm 2** Multi-step purification |
|---|---|
| **Input:** $x$: a test example; 
 $\qquad$ $T$: the number of attack steps 
 **Output:** $x_{\text{adv}}$: the adversarial example 
 1: $\delta \leftarrow 0$ 
 2: **for** $t = 1, 2, \ldots, T$ **do** 
 3: $\qquad \ell \leftarrow \mathcal{L}_{\text{cls}}((g \circ f)(x + \delta; \theta_{\text{enc}}, \theta_{\text{cls}}), y)$ 
 4: $\qquad \delta \leftarrow \delta + \gamma \, \text{sign}(\nabla_x \ell)$ 
 5: $\qquad \delta \leftarrow \min(\max(\delta, -\epsilon_{\text{adv}}), \epsilon_{\text{adv}})$ 
 6: $\qquad \delta \leftarrow \min(\max(x + \delta, 0), 1) - x$ 
 7: **end for** 
 8: $x_{\text{adv}} \leftarrow x + \delta$ | **Input:** $x$: a test example; 
 $\qquad$ $T$: the number of purification steps 
 **Output:** $x_{\text{pfy}}$: the purified example 
 1: $\delta \leftarrow 0$ 
 2: **for** $t = 1, 2, \ldots, T$ **do** 
 3: $\qquad \ell \leftarrow \mathcal{L}_{\text{aux}}((h \circ f)(x + \delta; \theta_{\text{enc}}, \theta_{\text{aux}}))$ 
 4: $\qquad \delta \leftarrow \delta - \gamma \, \text{sign}(\nabla_x \ell)$ 
 5: $\qquad \delta \leftarrow \min(\max(\delta, -\epsilon_{\text{pfy}}), \epsilon_{\text{pfy}})$ 
 6: $\qquad \delta \leftarrow \min(\max(x + \delta, 0), 1) - x$ 
 7: **end for** 
 8: $x_{\text{pfy}} \leftarrow x + \delta$ |

### 3.3 ONLINE PURIFICATION

Inspired by the PGD (Madry et al., 2017) attack (see Alg. 1), we propose a multi-step purifier (see Alg. 2) which can be seen as its inverse. In contrast to a PGD attack, which performs projected gradient *ascent* on the input in order to maximize the cross entropy loss $\mathcal{L}_{\text{cls}}$, the purifier performs projected gradient *descent* on the input in order to minimize the auxiliary loss $\mathcal{L}_{\text{aux}}$. The purifier achieves this goal by perturbing the adversarial examples, i.e. $\pi(x_{\text{adv}}) = x_{\text{adv}} + \delta$, while keeping the perturbation under a budget, i.e. $||\delta||_\infty \leq \epsilon_{\text{pfy}}$. Note that it is also plausible to use optimization-based algorithms in analogue to some $\ell_2$ adversaries such as CW (Carlini & Wagner, 2017), however this would require more steps of gradient descent at test-time.

Taking the bound into account, the final objective of the purifier is to minimize the following

$$\min_\delta \ \mathcal{L}_{\text{aux}}((h \circ f)(x_{\text{adv}} + \delta)) \ \text{ s.t. } ||\delta|| \leq \epsilon_{\text{pfy}}, \ x_{\text{adv}} + \delta \in [0, 1]. \tag{4}$$

For a multi-step purifier with step size $\gamma$, at each step we calculate

$$\delta_t = \delta_{t-1} + \gamma \, \text{sign}(\nabla_x \mathcal{L}_{\text{aux}}((h \circ f)(x_{\text{adv}} + \delta_{t-1}))). \tag{5}$$

For step size $\gamma = \epsilon_{\text{pfy}}$ and number of steps $T = 1$, the multi-step purifier becomes a single-step purifier. This is analogous to PGD degrading to FGSM (Goodfellow et al., 2014) when the step size of the adversary $\gamma = \epsilon_{\text{adv}}$ and the number of projected gradient ascent steps $T = 1$ in Alg. 1.

A remaining question is how to set $\epsilon_{\text{pfy}}$ when $\epsilon_{\text{adv}}$ is unknown. If $\epsilon_{\text{pfy}}$ is too small compared to the attack, it will not be sufficient to neutralize the adversarial perturbations. In the absence of knowledge of the attack $\epsilon_{\text{adv}}$, we can use the auxiliary loss as a proxy to set the appropriate $\epsilon_{\text{pfy}}$. In Figure 3 we plot the average auxiliary loss (green plot) of the purified examples for a range of $\epsilon_{\text{pfy}}$ values. The "elbows" of the auxiliary loss curves almost identify the unknown $\epsilon_{\text{adv}}$ in every case with slight over-estimation. This suggests that the value for which the auxiliary loss approximately stops decreasing is a good estimate of $\epsilon_{\text{adv}}$. Empirically, we find that using a slightly over-estimated $\epsilon_{\text{pfy}}$ benefits the accuracy after purification, similar to the claim by Song et al. (2018). This is because our network is trained with noisy examples and thus can handle the new noise introduced by purification. At test-time, we use the auxiliary loss to set $\epsilon_{\text{pfy}}$ in an online manner, by trying a range of values for $\epsilon_{\text{pfy}}$ and selecting the smallest one which minimizes the auxiliary loss for each *individual* example. In the experiment section we refer to the output of this selection procedure as $\epsilon_{\text{min-aux}}$. We also empirically find for each sample the $\epsilon_{\text{pfy}}$ that results in the best adversarial accuracy, denoted $\epsilon_{\text{oracle}}$ in the experiment section. This is an upper-bound on the performance SOAP can achieve.

### 3.4 SELF-SUPERVISED SIGNALS

Theoretically, any existing self-supervised objective can be used for purification. However, due to the nature of purification and also for the sake of efficiency, not every self-supervised task is suitable. A suitable auxiliary task should be sensitive to the representation shift caused by adversarial perturbation, differentiable with respect to the entire input, e.g. every pixel for an image, and also efficient in both train and test-time. In addition, note that certain tasks are naturally incompatible with certain datasets. For example, a rotation-based self-supervised task cannot work on a rotation-invariant dataset. In this paper, we exploit three types of self-supervised signals: data reconstruction, rotation prediction and label consistency.

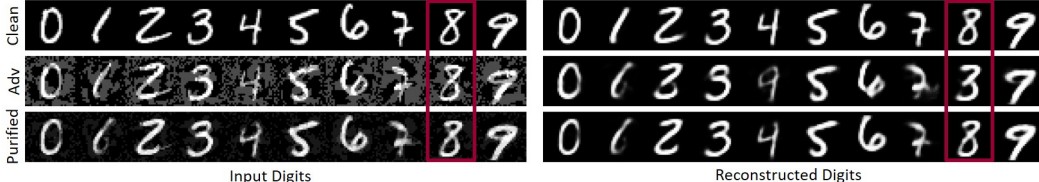

Figure 2: Input digits of the encoder (left) and output digits of the decoder (right). From top to bottom are the clean digits, adversarially perturbed digits and purified digits, respectively. Red rectangles: the adversary fools the model to incorrectly classify the perturbed digit 8 as a 3 and the purification corrects the perception back to an 8.

**Data reconstruction**   Data reconstruction (DR), including both deterministic data compression and probabilistic generative modeling, is probably one of the most natural forms of self-supervision. The latent representation, usually lying on a much lower dimensional space than the input space, is required to be comprehensive enough for the decoder to reconstruct the input data.

To perform data reconstruction, we use a decoder network as the auxiliary device $h$ and require it to reconstruct the input from the latent representation $z$. In order to better learn the underlying data manifold, as well as to increase robustness, the input is corrupted with additive Gaussian noise $\eta$ (and clipped) before fed into the encoder $f$. The auxiliary loss is the $\ell_2$ distance between examples and their noisy reconstruction via the autoencoder $h \circ f$:

$$\mathcal{L}_{\text{aux}} = ||x - (h \circ f)(x + \eta)||_2^2. \tag{6}$$

In Figure 2, we present the outputs of an autoencoder trained using Eq. (4), for clean, adversarial and purified inputs. The purification shifts the representation of the adversarial examples closer to their original class (for example, 2 4, 8 and 9). Note that SOAP does not use the output of the autoencoder as a defense, but rather uses the autoencoder loss to purify the input. We plot the autoencdoer output here as we consider it as providing insight to how the trained model 'sees' these samples.

**Rotation prediction**   Rotation prediction (RP), as an image self-supervised task, was proposed by Gidaris et al. (2018). The authors rotate the original images in a dataset by a certain degree, then use a simple classifier to predict the degree of rotation using high-level representation by a convolutional neural network. The rationale is that the learned representation has to be semantically meaningful for the classifier to predict the rotation successfully.

Following Gidaris et al. (2018), we make four copies of the image and rotate each of them by one of four degrees: $\Omega = \{0°, 90°, 180°, 270°\}$. The auxiliary task is a 4-way classification using representation $z = f(x)$, for which we use a simple linear classifier as the auxiliary device $h$. The auxiliary loss is the summation of 4-way classification cross entropy of each rotated copy

$$\mathcal{L}_{\text{aux}} = -\sum_{\omega \in \Omega} \log(h(f(x_\omega))_\omega) \tag{7}$$

where $x_\omega$ is a rotated input, and $h(\cdot)_\omega$ is the predictive probability of it being rotated by $\omega$. While the standard rotation prediction task works well for training, we found that it tends to under-estimate $\epsilon_{\text{pfy}}$ at test-time. Therefore, for purification we replace the cross entropy classification loss by the mean square error between predictive distributions and one-hot targets. This increases the difficulty of the rotation prediction task and leads to better robust accuracy.

**Label consistency**   The rationale of label consistency (LC) is that different data augmentations of the same sample should get consistent prediction from the network. This exact or similar concept is widely used in semi-supervised learning (Sajjadi et al., 2016; Laine & Aila, 2016), and also successfully applied in self-supervised contrastive learning (He et al., 2020; Chen et al., 2020b).

We adopt label consistency to perform purification. The auxiliary task here is to minimize the $\ell_2$ distance between two augmentations $a_1(x)$ and $a_2(x)$ of a given image $x$, in the logit space given by $g(\cdot)$. The auxiliary device of LC is the exact classifier, i.e. $h = g$, and the auxiliary loss

$$\mathcal{L}_{\text{aux}} = ||(g \circ f)(a_1(x)) - (g \circ f)(a_2(x))||_2^2. \tag{8}$$

## 4   EXPERIMENTS

We evaluate SOAP on the MNIST, CIFAR10 and CIFAR100 datasets following Madry et al. (2017).

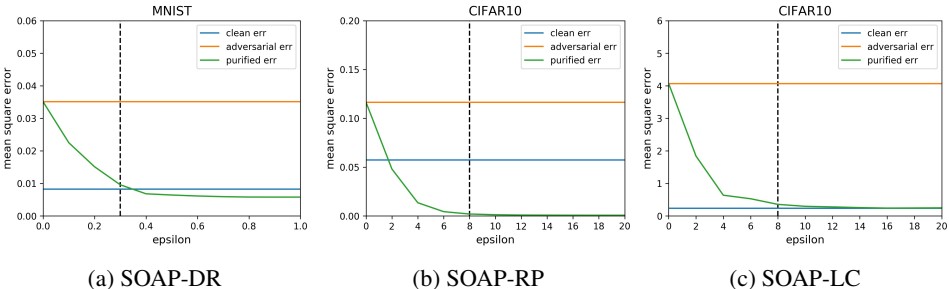

(a) SOAP-DR             (b) SOAP-RP             (c) SOAP-LC

Figure 3: Auxiliary loss vs. $\epsilon_{\text{pfy}}$. SOAP (green plot) reduces the high adversarial auxiliary loss (orange plot) to the low clean level (blue plot). The vertical dashed line is the value of $\epsilon_{\text{adv}}$. The trained models are FCN and ResNet-18 for MNIST and CIFAR10, respectively, with a PGD attack.

**MNIST (LeCun et al., 1998)** For MNIST, we evaluate our method on a fully-connected network (FCN) and a convolutional neural network (CNN). For the auxiliary task, we evaluate the efficacy of data reconstruction. For the FCN $g(\cdot)$ is a linear classifier and $h(\cdot)$ is a fully-connected decoder; for the CNN $g(\cdot)$ is a 2-layer MLP and $h(\cdot)$ is a convolutional decoder. The output of the decoder is squashed into the range of $[0, 1]$ by a sigmoid function. During training, the input digits are corrupted by an additive Gaussian noise ($\mu = 0, \sigma = 0.5$). At test-time, $\mathcal{L}_{\text{aux}}$ of the reconstruction is computed without input corruption. SOAP runs $T = 5$ iterations with step size $\gamma = 0.1$.

**CIFAR10 & CIFAR100 (Krizhevsky & Hinton, 2009)** For CIFAR10 and CIFAR100, we evaluate our method on a ResNet-18 (He et al., 2016) and a 10-widen Wide-ResNet-28 (Zagoruyko & Komodakis, 2016). For the auxiliary task, we evaluate rotation prediction and label consistency. To train on rotation prediction, each rotated copy is corrupted by an additive Gaussian noise ($\mu = 0, \sigma = 0.1$), encoded by $f(\cdot)$, and classified by a linear classifier $g(\cdot)$ for object recognition and by an auxiliary linear classifier $h(\cdot)$ for degree prediction. This results in a batch size 4 times larger than the original. At test-time, similar to DA, we compute $\mathcal{L}_{\text{aux}}$ on clean input images.

To train on label consistency, we augment the input images twice using a composition of random flipping, random cropping and additive Gaussian corruption ($\mu = 0, \sigma = 0.1$). Both of these two augmentations are used to train the classifier, therefore the batch size is twice as large as the original. At test-time, we use the input image as one copy and flip-crop the image to get another copy. Using the input image ensures that every pixel in the image is purified, and using definite flipping and cropping ensures there is enough difference between the input image and its augmentation. For both rotation prediction and label consistency, SOAP runs $T = 5$ iterations with step size $\gamma = 4/255$.

Note that we did not apply all auxiliary tasks on all datasets due to the compatibility issue mentioned in Sec. 3.4. DR is not suitable for CIFAR as reconstruction via an autoencdoer is typically challenging on more realistic image datasets. RP is naturally incompatible with MNIST because the digits 0, 1, and 8 are self-symmetric; and digits 6 and 9 are interchangeable with 180 degree rotation. Similarly, LC is also not appropriate for MNIST because common data augmentations such as flipping and cropping are less meaningful on digits.

### 4.1 WHITE-BOX ATTACKS

In Tables 1- 3 we compare SOAP against widely-used adversarial training (Goodfellow et al., 2014; Madry et al., 2017) and purification methods (Samangouei et al., 2018; Song et al., 2018) on a variety of $\ell_2$ and $\ell_\infty$ bounded attacks: FGSM, PGD, CW, and DeepFool (Moosavi-Dezfooli et al., 2016). For MNIST, both FGSM and PGD are $\ell_\infty$ bounded with $\epsilon_{\text{adv}} = 0.3$, and the PGD runs 40 iterations with a step size of $0.01$; CW and DeepFool are $\ell_2$ bounded with $\epsilon_{\text{adv}} = 4$. For CIFAR10, FGSM and PGD are $\ell_\infty$ bounded with $\epsilon_{\text{adv}} = 8/255$, and PGD runs 20 iterations with a step size of $2/255$; CW and DeepFool are $\ell_2$ bounded with $\epsilon_{\text{adv}} = 2$. For CW and DeepFool which are optimization-based, resulted attacks that exceed the bound are projected to the $\epsilon$-ball. We mark the best performance for each attack by an underlined and bold **value** and the second best by bold **value**. We do not mark out the oracle accuracy but it does serves as an empirical upper bound of purification.

For MNIST, SOAP-DR has great advantages over FGSM and PGD adversarial training on all attacks when the model has small capacity (FCN). This is because adversarial training typically requires a large parameter set to learn a complex decision boundary while our method does not have this

constraint. When using a larger CNN, SOAP outperforms FGSM adversarial training and comes close to Defense-GAN and PGD adversarial training on $\ell_\infty$ attacks. SOAP also achieves better clean accuracy compared with all other methods.

Note that FGSM AT achieves better accuracy under FGSM attacks than when there is no attack for the large capacity networks. This is due to the label leaking effect (Kurakin et al., 2016): the model learns to classify examples from their perturbations rather than the examples themselves.

Table 1: MNIST Results

| Method | FCN | | | | | CNN | | | | |
|---|---|---|---|---|---|---|---|---|---|---|
| | No Atk | FGSM | PGD | CW | DF | No Atk | FGSM | PGD | CW | DF |
| No Def | **98.10** | 16.87 | 0.49 | 0.01 | 1.40 | **99.15** | 1.49 | 0.00 | 0.00 | 0.69 |
| FGSM AT | 79.76 | **80.57** | 2.95 | 6.22 | 17.24 | 98.78 | **99.50** | 33.70 | 0.02 | 6.16 |
| PGD AT | 76.82 | 60.70 | 57.07 | 31.68 | 13.82 | 98.97 | **96.38** | **93.22** | 90.31 | 75.55 |
| Defense-GAN | 95.84 | **79.30** | **84.10** | **95.07** | **95.29** | 95.92 | 90.30 | **91.93** | **95.82** | **95.68** |
| SOAP-DR | | | | | | | | | | |
| $\epsilon_{\text{pfy}} = 0$ | 97.57 | 29.15 | 0.58 | 0.25 | 2.32 | **99.04** | 65.35 | 27.54 | 0.35 | 0.69 |
| $\epsilon_{\text{pfy}} = \epsilon_{\text{min-aux}}$ | **97.56** | 66.85 | **61.88** | **86.81** | **87.02** | 98.94 | 87.78 | 84.92 | 74.61 | **81.27** |
| $\epsilon_{\text{pfy}} = \epsilon_{\text{oracle}}*$ | 98.93 | 69.21 | 64.76 | 97.88 | 97.97 | 99.42 | 89.40 | 86.62 | 98.44 | 98.47 |

Table 2: CIFAR-10 results

| Method | ResNet-18 | | | | | Wide-ResNet-28 | | | | |
|---|---|---|---|---|---|---|---|---|---|---|
| | No Atk | FGSM | PGD | CW | DF | No Atk | FGSM | PGD | CW | DF |
| No Def | **90.54** | 15.42 | 0.00 | 0.00 | 6.26 | **95.13** | 14.82 | 0.00 | 0.00 | 3.28 |
| FGSM AT | 72.73 | 44.16 | 37.40 | 2.69 | 24.58 | 72.20 | **91.63** | 0.01 | 0.00 | 14.41 |
| PGD AT | 74.23 | **47.43** | **42.11** | 3.14 | 25.84 | 85.92 | 51.58 | 41.50 | 2.06 | 24.08 |
| Pixel-Defend | 79.00 | 39.85 | 29.89 | **76.47** | **76.89** | 83.68 | 41.37 | 39.00 | 79.30 | **79.61** |
| SOAP-RP | | | | | | | | | | |
| $\epsilon_{\text{pfy}} = 0$ | 73.64 | 5.77 | 0.47 | 0.00 | 13.65 | 88.68 | 30.21 | 8.52 | 0.08 | 10.67 |
| $\epsilon_{\text{pfy}} = \epsilon_{\text{min-aux}}$ | 71.97 | 35.80 | 38.53 | 68.22 | 68.44 | 90.94 | 51.11 | **51.90** | **83.03** | **82.50** |
| $\epsilon_{\text{pfy}} = \epsilon_{\text{oracle}}*$ | 87.57 | 37.60 | 39.40 | 79.80 | 84.34 | 95.55 | 52.69 | 52.61 | 86.99 | 90.49 |
| SOAP-LC | | | | | | | | | | |
| $\epsilon_{\text{pfy}} = 0$ | 86.36 | 22.81 | 0.15 | 0.00 | 8.52 | **93.40** | 59.23 | 3.55 | 0.01 | 46.98 |
| $\epsilon_{\text{pfy}} = \epsilon_{\text{min-aux}}$ | **84.07** | **51.02** | **51.42** | 73.95 | 74.79 | 91.89 | **64.83** | **53.58** | 80.33 | 60.56 |
| $\epsilon_{\text{pfy}} = \epsilon_{\text{oracle}}*$ | 94.06 | 59.45 | 62.29 | 86.94 | 88.88 | 96.93 | 71.85 | 63.10 | 88.96 | 73.66 |

Table 3: CIFAR-100 results

| Method | ResNet-18 | | | | | Wide-ResNet-28 | | | | |
|---|---|---|---|---|---|---|---|---|---|---|
| | No Atk | FGSM | PGD | CW | DF | No Atk | FGSM | PGD | CW | DF |
| No Def | **65.56** | 3.81 | 0.01 | 0.00 | 12.30 | **78.16** | 13.76 | 0.06 | 0.01 | 9.05 |
| FGSM AT | 44.35 | 20.30 | 17.41 | 4.23 | 18.15 | 46.45 | **88.24** | 0.15 | 0.00 | 13.40 |
| PGD AT | 42.15 | **21.92** | **20.04** | 3.57 | 17.90 | 62.71 | 28.15 | 21.34 | 0.65 | 16.57 |
| SOAP-RP | | | | | | | | | | |
| $\epsilon_{\text{pfy}} = 0$ | 40.47 | 2.53 | 0.45 | 0.03 | 11.89 | 60.33 | 13.30 | 4.65 | 0.09 | 12.19 |
| $\epsilon_{\text{pfy}} = \epsilon_{\text{min-aux}}$ | 35.21 | 11.65 | 11.73 | **32.97** | **33.51** | 60.80 | 22.25 | **22.00** | 54.11 | **54.70** |
| $\epsilon_{\text{pfy}} = \epsilon_{\text{oracle}}*$ | 45.57 | 12.44 | 12.04 | 41.13 | 46.51 | 72.03 | 24.42 | 24.19 | 63.04 | 67.86 |
| SOAP-LC | | | | | | | | | | |
| $\epsilon_{\text{pfy}} = 0$ | **57.86** | 6.11 | 0.01 | 0.01 | 12.72 | **74.04** | 16.46 | 0.49 | 0.00 | 9.65 |
| $\epsilon_{\text{pfy}} = \epsilon_{\text{min-aux}}$ | 52.91 | **22.93** | **27.55** | **50.26** | **50.57** | 61.01 | **31.40** | **37.53** | **56.09** | 53.79 |
| $\epsilon_{\text{pfy}} = \epsilon_{\text{oracle}}*$ | 69.99 | 27.52 | 31.82 | 62.87 | 68.65 | 82.74 | 37.56 | 47.07 | 71.19 | 73.39 |

For CIFAR10, on ResNet-18 SOAP-RP beats Pixel-Defend on all attacks except for FGSM and beats PGD adversarial training on $\ell_2$ attacks; on Wide-ResNet-28 it performs superiorly or equivalently

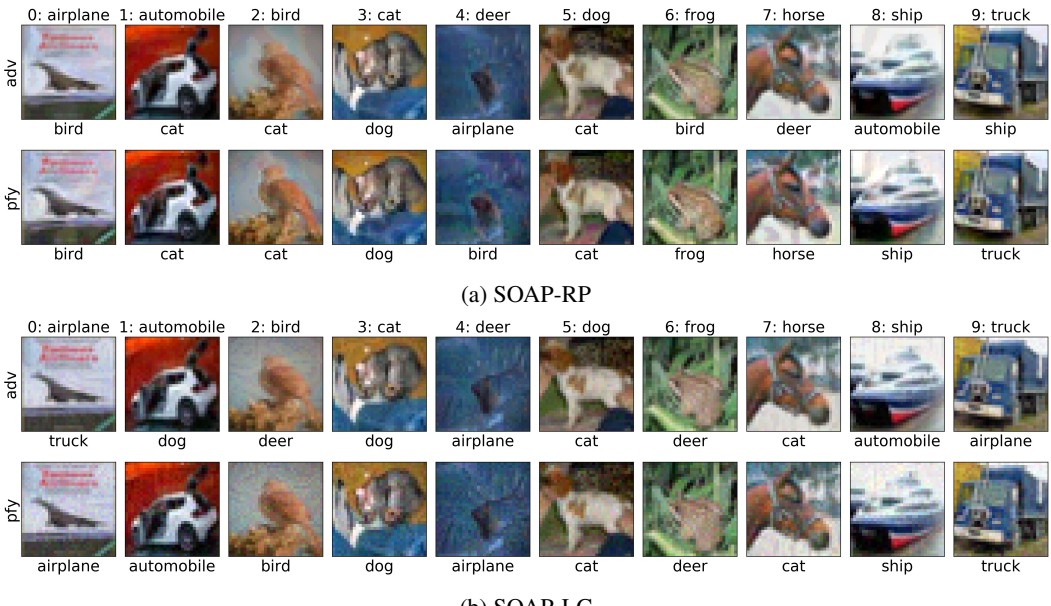

Figure 4: Adversarial and purified CIFAR10 examples by SOAP with Wide-ResNet-28 under PGD attacks. True classes are shown on the top of each column and the model predictions are shown under each image.

against other methods on all attacks. SOAP-LC achieves superior accuracy compared with other methods, where the capacity is either small or large. Note that we choose Pixel-Defend as our purification baseline since Defense-GAN does not work on CIFAR10. Specifically, our method achieves over $50\%$ accuracy under strong PGD attack, which is $10\%$ higher than PGD adversarial training. SOAP also exhibits great advantages over adversarial training on the $\ell_2$ attacks. Also, compared with vanilla training ('No Def') the multi-task training of SOAP improves robustness without purification ($\epsilon_{\text{pfy}} = 0$), which is similar on MNIST. Examples are shown in Figure 4.

For CIFAR100, SOAP shows similar advantages over other methods. SOAP-RP beats PGD adversarial training on PGD attacks when using large Wide-ResNet-28 model and on $\ell_2$ attacks in all cases; SOAP-LC again achieves superior accuracy compared with all other methods, where the capacity is either small or large.

Our results demonstrate that SOAP is effective under both $\ell_\infty$ and $\ell_2$ bounded attacks, as opposed to adversarial training which only defends effectively against $\ell_2$ attacks for MNIST with a CNN. This implies that while the formulation of the purification in Eq. (4) mirrors an $\ell_\infty$ bounded attack, our defense is not restricted to this specific type of attack, and the bound in Eq. (4) serves merely as a constraint on the purification perturbation rather than a-prior knowledge of the attack.

**Auxiliary-aware attacks**   Previously, we focus on standard adversaries which only rely on the classification objectives. A natural question is: can an adversary easily find a stronger attack given the knowledge of our purification defense? In this section, we introduce a more 'complete' white-box adversary which is aware of the purification method, and show that it is not straightforward to attack SOAP even with the knowledge of the auxiliary task used for purification.

In contrast to canonical adversaries, here we consider adversaries that jointly optimize the cross entropy loss and the auxiliary loss with respect to the input. As SOAP aims to minimize the auxiliary loss, the auxiliary-aware adversaries maximize the cross entropy loss while also minimizing the auxiliary loss at the same time. The intuition behind this is that the auxiliary-aware adversaries try to find the auxiliary task "on-manifold" (Stutz et al., 2019) examples that can fool the classifier. The auxiliary-aware adversaries perform gradient ascent on the following combined objective

$$\max_\theta \{\mathcal{L}_{\text{cls}}(f(x), y; \theta_{\text{enc}}, \theta_{\text{cls}}) - \beta \mathcal{L}_{\text{aux}}(g(x; \theta_{\text{enc}}, \theta_{\text{aux}}))\}, \tag{9}$$

where $\beta$ is a trade-off parameter between the cross entropy loss and the auxiliary loss. An auxiliary-aware adversary degrades to a canonical one when $\beta = 0$ in the combined objective.

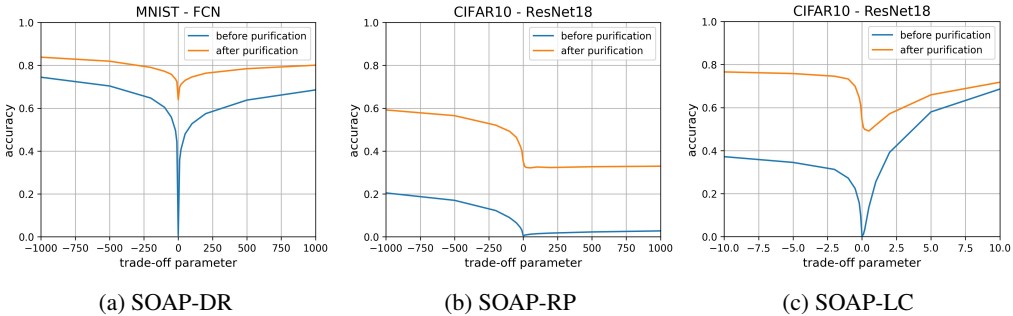

|          (a) SOAP-DR          |          (b) SOAP-RP          |          (c) SOAP-LC          |

Figure 5: Purification against auxiliary-aware PGD attacks. Plots are classification accuracy before (blue) and after (orange) purification.

As shown in Figure 5, an adversary cannot benefit from the knowledge of the defense in a straight-forward way. When the trade-off parameter $\beta$ is negative (i.e. the adversary is attacking the auxiliary device as well), the attacks are weakened (blue plot) and purification based on all three auxiliaries achieves better robust accuracy (orange plot) as the amplitude of $\beta$ increases. When $\beta$ is positive, the accuracy of SOAP using data reconstruction and label consistency increases with $\beta$. The reason for this is that the auxiliary component of the adapted attacks obfuscates the cross entropy gradient, and thus weakens canonical attacks. The accuracy of rotation prediction stays stable as $\beta$ varies, i.e. it is more sensitive to this kind of attack compared to the other tasks.

## 4.2    BLACK-BOX ATTACKS

Table 4 compares SOAP-DR with adversarial training against FGSM black-box attacks (Papernot et al., 2017). Following their approach, we let white-box adversaries, e.g. FGSM, attack a substitute model, with potentially different architecture, to generate the black-box adversarial examples for the target model. The substitute model is trained on a limited set of 150 test images unseen by the target model. These images are further labeled by the target model and augmented using a Jacobian-based method. SOAP significantly out-performs adversarial training on FCN; for CNN it out-performs FGSM adversarial training and comes close to PGD adversarial training.

Table 4: MNIST Black-box Results

| Target |  | FCN |  |  | CNN |  |
|---|---|---|---|---|---|---|
| Substitute | No Atk | FCN | CNN | No Atk | FCN | CNN |
| No Def | **98.10** | 25.45 | 39.10 | **99.15** | 49.49 | 49.25 |
| FGSM AT | 79.76 | 40.88 | 58.74 | 98.78 | 93.62 | 96.52 |
| PGD AT | 76.82 | 62.87 | 69.07 | 98.97 | **97.79** | **98.09** |
| SOAP-DR |  |  |  |  |  |  |
| $\epsilon_{\text{pfy}} = 0$ | **97.57** | 78.52 | 92.72 | **99.04** | 95.25 | 97.43 |
| $\epsilon_{\text{pfy}} = \epsilon_{\text{min-aux}}$ | 97.56 | **90.35** | **94.51** | 98.94 | **96.02** | **97.80** |
| $\epsilon_{\text{pfy}} = \epsilon_{\text{oracle}}\ast$ | 98.93 | 94.34 | 97.33 | 99.42 | 98.12 | 98.81 |

## 5    CONCLUSION

In this paper, we introduced SOAP: using self-supervision to perform test-time purification as an online defense against adversarial attacks. During training, the model learns a clean data manifold through joint optimization of the cross entropy loss for classification and a label-independent auxiliary loss for purification. At test-time, a purifier counters adversarial perturbation through projected gradient descent of the auxiliary loss with respect to the input. SOAP is consistently competitive across different network capacities as well as different datasets. We also show that even with knowledge of the self supervised task, adversaries do not gain an advantage over SOAP. While in this paper we only explore how SOAP performs on images, our purification approach can be extended to any data format with suitable self-supervised signals. We hope this paper can inspire future exploration on a broader range of self-supervised signals for adversarial purification.

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

## A  APPENDIX

### A.1  ILLUSTRATION OF SELF-SUPERVISED TASKS

For those readers who are not familiar with self-supervised representation learning, we detailed the self-supervised tasks here. As data reconstruction is straight-forward to understand, we simply illustrate rotation prediction and label consistency in Figure 6. As shown on the left-hand side, to perform rotation prediction, we duplicate a input image to 4 copies and rotate them by one of the four degrees. We then use the auxiliary classifier to predict the rotation of each copy. For the label consistency auxiliary task on the right-hand side, we duplicate a input image to 2 copies and apply separate data augmentation to each of them. The left copy is augmented with random cropping and the right copy is augmented with random cropping as well as horizontal flipping. We require the predictive distributions of these 2 augmented copies to be close.

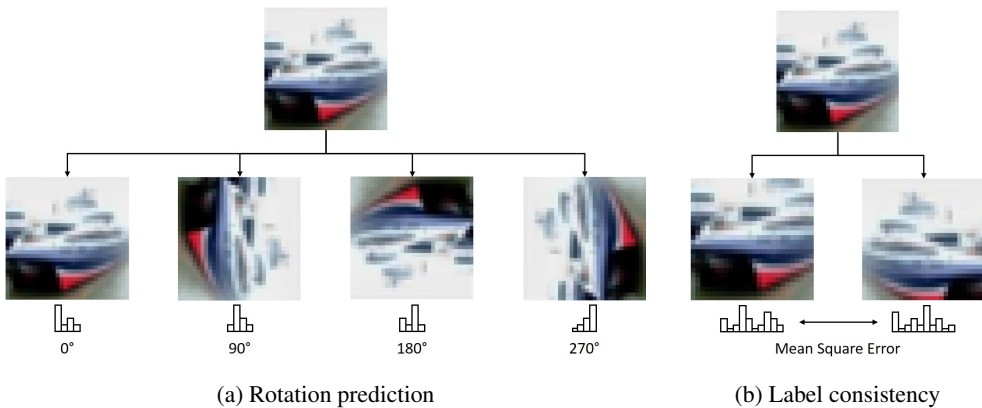

(a) Rotation prediction         (b) Label consistency

Figure 6: An illustration of auxiliary self-supervised tasks.

### A.2  HYPER-PARAMETERS OF THE PURIFIER

Beyond $\epsilon_{\text{pfy}}$, the two additional hyper-parameters of SOAP are the step size of the purifier $\gamma$, and the number of iterations performed by the purifier $T$. The selection of these hyper-parameters is important for the efficacy of purification. A step size that is too small or a number of iterations that is too large can cause the purifier to get stuck in a local minimum neighboring the perturbed example. This is confirmed by our empirical finding that using a relatively large step size for a small number of iterations is better than using a relatively small step size for a large number of iterations. Although a large step size makes it hard to get the cleanest purified examples, this drawback is compensated by adding noise in training. Training the model on corrupted examples makes the model robust to the residual noise left by purification.

### A.3  TRAINING DETAILS

Table 5: Basic modules & specifics

| Module | Specifics |
|---|---|
| Conv($m, k \times k, s$) | 2-D convolutional layer with $m$ feature maps, $k \times k$ kernel size, and stride $s$ on both directions |
| Maxpool($s$) | 2-D maxpooling layer with stride $s$ on both directions |
| FC($m$) | Fully-connected layer with $m$ outputs |
| ReLU | Rectified linear unit activation |
| BN | 1-D or 2-D batch normalization |
| Dropout($p$) | Dropout layer with probability $p$ |
| ShortCut | Residual addition that bypasses the basic block |

Table 6: ResNet basic blocks

| Block($m, k \times k, s$) | Wide-Block($m, k \times k, s, p$) |
|---|---|
| Conv($m, k \times k, s$) | Conv($m \times 10, k \times k, 1$) |
| BN | Dropout($p$) |
| ReLU | BN |
| Conv($m, k \times k, 1$) | ReLU |
| BN | Conv($m \times 10, k \times k, s$) |
| ShortCut | ShortCut |
| ReLU | BN |
| | ReLU |

Table 7: Architectures

| FCN | CNN | ResNet-18 | Wide-ResNet-28 |
|---|---|---|---|
| FC(256) | Conv($32, 3 \times 3, 2$) | Conv($16, 3 \times 3, 1$) | Conv($16, 3 \times 3, 1$) |
| ReLU | ReLU | BN | BN |
| Dropout(0.5) | Conv($64, 3 \times 3, 2$) | ReLU | ReLU |
| FC(128) | ReLU | Block($16, 3 \times 3, 1$) $\times 2$ | Wide-Block($16, 3 \times 3, 1, 0.3$) $\times 4$ |
| ReLU | BN | Block($32, 3 \times 3, 2$) | Wide-Block($32, 3 \times 3, 2, 0.3$) |
| Dropout(0.5) | FC(128) | Block($32, 3 \times 3, 1$) | Wide-Block($32, 3 \times 3, 1, 0.3$) $\times 3$ |
| FC(10) | ReLU | Block($64, 3 \times 3, 2$) | Wide-Block($64, 3 \times 3, 2, 0.3$) |
| | BN | Block($64, 3 \times 3, 1$) | Wide-Block($64, 3 \times 3, 1, 0.3$) |
| | FC(10) | FC(10) | FC(10) |

The architectures of the networks and training details are described as followed. Table 5 describes the basic modules and their specifics, and Table 6 describes the low-level building blocks of residual networks. Full architectures of the networks are listed in Table 7.

For MNIST, we evaluate on a 2 hidden layer FCN and a CNN which has the same architecture as in Madry et al. (2017). FCN is trained for 100 epochs with an initial learning rate of $0.01$ and CNN for 200 epochs with an initial learning rate of $0.1$ using SGD. The learning rate is decreased 10 times at halfway in both cases. The batch size is 128. In both FGSM and PGD adversarial training, the adversaries are $l_\infty$ bounded with $\epsilon_{\text{adv}} = 0.3$. For PGD adversarial training, the adversary runs 40 steps of projected gradient descent with a step size of $0.01$. To train SOAP, the trade-off parameter $\alpha$ in Eq. (1) is 100.

For CIFAR10, we evaluate our method on a regular residual network ResNet-18 and a 10-widen residual network Wide-ResNet-28-10. Both networks are trained for 200 epochs using a SGD optimizer. The initial learning rate is 0.1, which is decreased by a factor of 0.1 at the 100 and 150 epochs. We use random crop and random horizontal flipping for data augmentation on CIFAR10. $\epsilon_{\text{adv}} = 8/255$ for both FGSM and PGD adversarial training. For PGD adversarial training, the adversary runs 7 steps of projected gradient descent with a step size of $2/255$. The trade-off parameter of SOAP for rotation prediction and label consistency $\alpha$ are 0.5 and 1 respectively.

While we implement adversarial training and SOAP ourselves, we use authors' implementation for both Defense-GAN and Pixel-Defend. Notice that our white-box Defense-GAN accuracy is lower than the accuracy reported in (Samangouei et al., 2018). Part of the reason is the difference in architecture and training scheme, but we are still not able to replicate their accuracy using the exact same architecture following their instructions. Nonetheless, our results are close to (Hwang et al., 2019) where the authors also reported lower accuracy.

## A.4 Training efficiency

The comparison of training efficiency between SOAP and other methods is shown in Figure 7. To measure the training complexity, we run each training method for 30 epochs on a single Nvidia Quadro RTX 8000 GPU, and report the average epoch time consumption. When the network capacity is small, the training complexity of SOAP is close to FGSM adversarial training and much lower than PGD adversarial training. When the network capacity is large, the training complexity

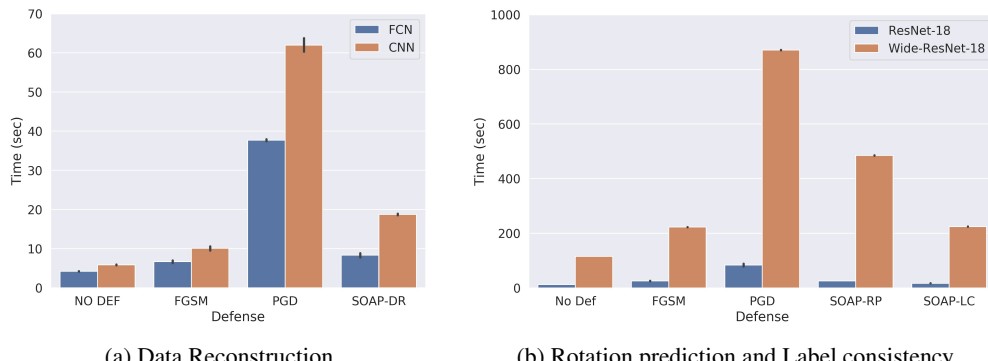

(a) Data Reconstruction          (b) Rotation prediction and Label consistency

Figure 7: Comparison of training efficiency between SOAP, vanilla training ('No Def') and adversarial training (FGSM and PGD). The y-axis is the average time consumption of 30 training epochs.

of SOAP is higher than FGSM adversarial training but still significantly lower than PGD adversarial training.

Note that it is hard to compare with other purification methods because they are typically trained in 2 stages, the training of the classifier and the training of another purifier such as a GAN. While the training of those purifiers is typically difficult and intense, SOAP does not suffer from this limitation as the encoder is shared between the main classification network and the purifier. Therefore it is reasonable to claim that SOAP is more efficient than other purification methods.

## A.5    PURIFIED EXAMPLES

We have shown some PGD examples of adversarial images purified images by SOAP in Figure 4. In Figures 8-10 we present some examples of every attack. The adversary is shown on the top of each column. The network prediction is shown at the bottom of each example.

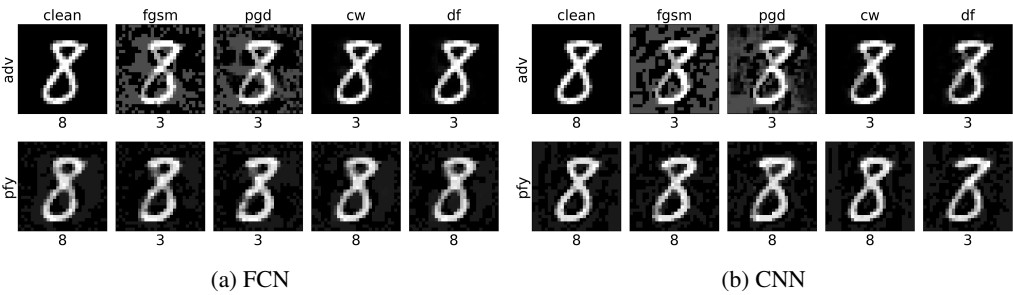

(a) FCN          (b) CNN

Figure 8: MNIST examples with data reconstruction.

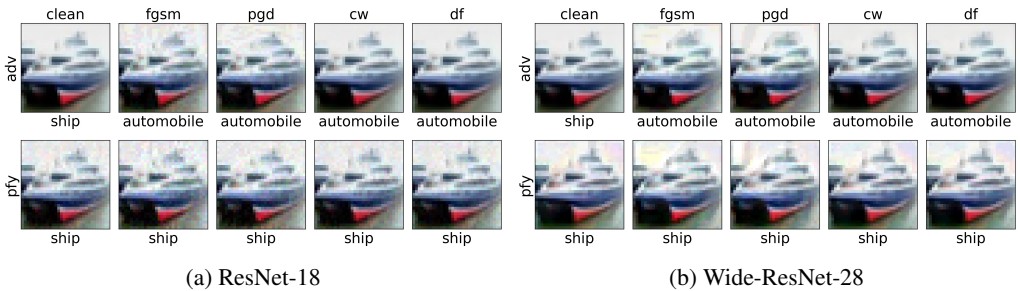

(a) ResNet-18          (b) Wide-ResNet-28

Figure 9: CIFAR10 examples with rotation prediction.

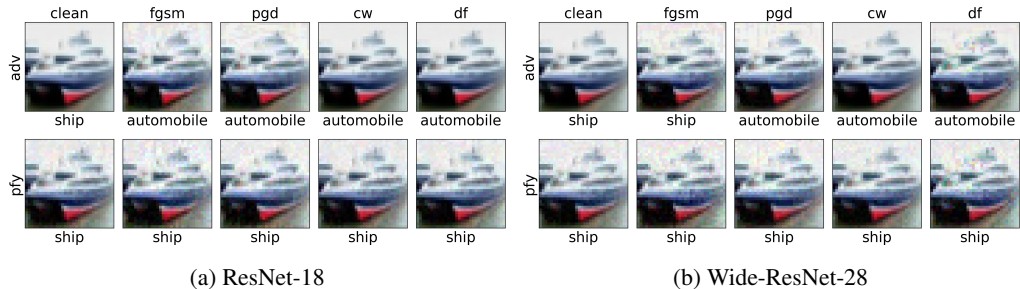

Figure 10: CIFAR10 examples with label consistency.

## A.6 SUCCESS RATE OF $\ell_2$ ATTACKS

To provide more details about the robustness under $\ell_2$ attacks, in Tables 8 and 9 we report the success rate of generating $\ell_2$ attacks. For the MNIST dataset, generating an $\ell_2$ attack is considered a success if the $\ell_2$ norm of the perturbation is smaller than 4; for the CIFAR10 dataset, the bound is set as 2. PGD adversarial training typically results in low success rate compared with other methods; SOAP, on the other hand, is typically "easy" to attack before purification because there is no adversarial training involved. Notice that the Wide-ResNet-28 model trained on label consistency is robust to the DeepFool attack even before purification. This explains why SOAP-LC robust accuracy in Table 2 is relatively low because its DeepFool perturbations are larger than other cases.

Table 8: MNIST $\ell_2$ attacks success rate

| Attack | Architecture | No Def | FGSM AT | PGD AT | SOAP-DR |
|--------|-------------|--------|---------|--------|---------|
| CW | FCN | 100.00 | 99.98 | 99.98 | 99.78 |
|    | CNN | 100.00 | 100.00 | 100.00 | 99.82 |
| DF | FCN | 99.88 | 94.90 | 99.10 | 99.53 |
|    | CNN | 99.92 | 100.00 | 18.97 | 95.33 |

Table 9: CIFAR10 $\ell_2$ attacks success rate

| Attack | Architecture | No Def | FGSM AT | PGD AT | SOAP-RP | SOAP-LC |
|--------|-------------|--------|---------|--------|---------|---------|
| CW | ResNet-18 | 100.00 | 97.31 | 96.88 | 100.00 | 100.00 |
|    | Wide-ResNet-28 | 100.00 | 100.00 | 97.95 | 99.92 | 99.99 |
| DF | ResNet-18 | 100.00 | 88.20 | 87.01 | 99.99 | 99.85 |
|    | Wide-ResNet-28 | 100.00 | 100.00 | 83.68 | 96.31 | 28.22 |

