# OpenReview forum: "Online Adversarial Purification based on Self-supervised Learning"
_ICLR.cc/2021/Conference — ICLR 2021 Poster_

### Official Review · AnonReviewer2 · 2020-10-26
**Official Blind Review #2**

**Rating:** 7
**Confidence:** 5

**Review:**

[Summary]
Online defenses of adversarial examples is an old topic: Given an input x (potentially adversarially perturbed) at test time, we want to sanitize x to get x', on which the trained classifier $g \circ f$ gives the correct answer. This paper proposes a new architecture for online defenses via self supervision. There are two new things in the proposal:

1. There is an explicit representation function f, namely the classifier is decomposed into $g \circ f$. And the auxiliary self-supervised component h works on the same representation. This thus creates a Y-shape architecture that is "syntactically" similar to the training structure in unsupervised domain adaptation (e.g., domain adversarial neural networks). This architecture for online defense seems new (as far as I know).

2. The paper leverages an interesting hypothesis that for a common f, a large classification loss happens if and only if a large self-supervision loss happens. And this paper provides solid evidence to justify this -- namely in Section 4.1 (auxiliary-aware attacks), it evaluates the defense against an adversary that is aware of h, in order to create adversarial examples that explicitly breaks the hypothesis (i.e. large classification loss but small self-supervision loss).

3. For the experiments -- the paper trained f, g, and h under Gaussian corruptions, and indeed found that this online purification strategy provides robustness under adversarial perturbations, even for auxiliary-aware attacks, which is interesting.

[Assessment]
1. My first worry is that the performance of the defense is still much worse than the performance from direct adversarial training (for example, check the MNIST numbers). For example, under PGD, on CNN architecture we can achieve 80%ish accuracy. Note that for MNST, a simple discretization can already achieve almost-perfect accuracy. This is especially the case if we consider auxiliary-aware attacks.

2. Following (1), what worries me more is that online-purification still needs to be aware of the attack type. Namely if one looks into equation (4), the objective has encoded norm-based attacks within it. This makes the results less interesting.

3. All in all, my major doubt is what is really the benefit of reduced training complexity if we cannot achieve better robustness, and also the defense still needs to be fully aware of the attack type? For these reasons, I vote for a weak reject.

[Questions]
1. Why do we need to know the results for FCN (fully connected networks)?

2. I am not sure the numbers reported for adversarial training match the state of the art reported in the MNIST challenge leaderboard: https://github.com/MadryLab/mnist_challenge. There the SOTA MNIST model always has >88% accuracy (so I am a bit skeptical about DF can bring down the accuracy to 78% for PGD AT). Also, how about applying those attacks for the self-supervision defense? (that's an additional request). Similarly, for CIFAR10, as shown by https://github.com/MadryLab/cifar10_challenge, PGD AT is never under 43%, but in Table 2, the robust accuracy is only 2% under CW attack. This is suspicious.

[Post rebuttal]

After more discussion and reading through the revision, I think this is a good paper and will be useful to the community for an instance of test-time defenses.

---

> ### Author Response · Authors · 2020-11-23
> **Official Reply to Reviewer 2**
>
> We thank you for your valuable feedback on our paper. We would like to address your concerns accordingly:
>
> 1. By including the FCN results, we show that SOAP works for various architectures, and is not specific to CNNs. Though real-world visual understanding algorithms almost always rely on CNNs, the paradigm of SOAP is not specific to images and can easily generalize to other data formats, such as text and graphs. FCN or other architectures can be useful in those scenarios, combined with an appropriate self-supervised task.
> 2. The reason that our CW results seem low compared to the results posted on the leaderboard is that the two attacks are actually different despite bearing the same name. Specifically, our CW attack is the optimization-based l_2 bounded (as is DF) version from the original paper (please see Sec. 4.1), while the leaderboard’s CW attack is the PGD-based l_inf bounded version from the PGD paper. For optimization-based attacks, the accuracy of any classifier can always go down to 0 if the bound is large enough. We apologize for the lack of clarity.
>
> In response to some other concerns:
>
> 1. While it’s true that SOAP does not out-perform PGD AT with CNNs on MNIST (we actually close the gap by tuning the CNNs more carefully), SOAP does out-perform PGD and FGSM AT on MNIST when the model capacity is low, and on CIFAR10 and CIFAR100 in general (which are more realistic image datasets). SOAP also achieves very competitive accuracies on MNIST on black-box attacks.
> We also want to mention that, though not revealed in the paper, our experiments show that using uniform noises rather than Gaussian noises during training can improve the PGD accuracy of SOAP-DR to 90.13% with CNNs on MNIST. We choose to use Gaussian noises for overall performance on all considered attacks.
> 2. Our defense is not based on knowledge of the attack. Rather we need to constrain the form of our defense (the perturbation used to purify the image) in order to solve the optimization problem. While our defense does parallel the l_inf bounded attack, note that in our experiments we show it performs well on both l_inf bounded attacks (FGSM, PGD) and l_2 bounded attacks (CW, DF). This is in contrast to AT which uses knowledge of the attack type (l_inf) and performs poorly on l_2 attacks. Although our purification is based on l_inf-norm perturbations in Eq. 4 (because the purifier needs some budget), we show that SOAP can defend at least against l_inf and l_2 and potentially more. We have further clarified this in Sec. 3.3.
> 3. As above, our defense is not aware of the attack and we do obtain better accuracy on more difficult datasets than MNIST, e.g. CIFAR10 and CIFAR100. Note that we significantly out-perform AT on l_2 attacks.

---

> > ### Comment · AnonReviewer2 · 2020-11-23
> > **Quick replies on the "attack agnostic" argument**
> >
> > Thanks for the update. The fact that it can defend against $\ell_2$ attacks, with only $\ell_\infty$ training in place, is a good point.
> >
> > To push back a little, while I acknowledge that I have made very specific words of "knowing the attack type", what was (also) in my mind is that this whole process (note that the defense is actually training+purification, which are two algorithms), we are still doing adversarial training (and at least now, the training does not jump out of the norm-based adv training). So to certain extent from the empirical results, it is more "agnostic", but it is not the "real agnostic" in my mind (at least for the current evidence, maybe this is actually too much to hope for). For example, how about using your $\ell_\infty$-based defense to defend against rotation-based attacks (even just transfer attacks?) Nevertheless, I stress again that like the empirical evidence on defending $\ell_2$ attacks.
> >
> > Also, are you aware of any **theoretical evidence** why in this case training with $\ell_\infty$ can give $\ell_2$ robustness as well? I am asking this because adv training has a well-justified theoretical evidence that on the 1-layer network it becomes large-margin training (moreover, see this paper https://arxiv.org/abs/2005.10190 for more theory). I worry a bit about the robustness on $\ell_2$ will be brittle without any "bullet-proof" evidence (I know this is challenging, but I want to put this here).
> >
> > Finally, for this submission, I will need to talk to other reviewers in the post-rebuttal period, to finally decide my score. Thanks.

---

> > > ### Author Response · Authors · 2020-11-23
> > > **Further Clarification on the Training Strategy**
> > >
> > > Thank you for your response. We would like to further clarify that our defense strategy is purely online and does not rely on adversarial examples during training. Note that the training objective is Equation.2 where x is a natural example (corrupted by Gaussian noises), and we minimize a joint classification and auxiliary loss. No adversarial training is used in this.
> > > At test-time, we perform online purification based on Equation.4, which finds an l_inf bounded “correction” to the input image, again without knowledge of the attack type or its strength. We assume this performs well on l_2 attacks because the bounded purification perturbation is strong enough and the model is trained with noises to deal with residuals of imperfect purification.
> > > We agree that combining adversarial training is a promising direction to explore, but at the current stage we found that, with online/test-time purification, doing auxiliary/multi-task training on noisy natural examples alone already gives us decent results. This is also for the sake of efficiency - SOAP is more efficient during training than traditional adversarial training since it does not perturb examples adversarially.

---

> > > > ### Comment · AnonReviewer2 · 2020-11-23
> > > > **You are right, apologies for the confusions**
> > > >
> > > > I have a wrong memory about (2) and (3) (being a while after my last read). Yes objective (2) is natural training and only objective (3) (which is the online part) at the test-time has an adversarial training form. But isn't (3) a form of adversarial training? (Algorithm 1 and 2 are also adversarial training too). So the main evaluation is still "norm bounded adversarial training" against "norm bounded attacks". Can you try to defend against rotation-based attacks? I think that will demonstrate more surprising "attack agnostic".
> > > >
> > > > I think we are on the same page, but maybe I take a different perspective about what does it by "attack agnostic" (but I agree that your point about $\ell_2$ attacks is interesting).

---

> ### Comment · AnonReviewer2 · 2020-11-24
> **Update my score and rationales**
>
> Thanks the authors. I have updated my score to 7. For two reasons: (1) This paper has provided clear and reasonable evidence for robustness in the white-box case. And (2) It may have a potential to provide the first example of adversarial robustness (that actually works in the white-box model) where at the training time we do not need explicit adversarial training. In that sense, I think either confirming or refuting the claim is of interest to the community.

---

### Official Review · AnonReviewer3 · 2020-10-28
**Interesting method and results, I am missing results on larger and more complex datasets.**

**Rating:** 7
**Confidence:** 3

**Review:**

Summary: The paper introduces a defence for adversarial attack based on minimising a self-supervised loss on the test examples. Authors work under the assumption that minimising the self-supervised loss would be equivalent to minimising the supervised loss (to which they don't have access at test time). Authors evaluate their method on MNIST and CIFAR.

Strengths:
- The paper address the important topic of adversarial defence. Given the numerous adversarial attacks that have appeared in the last years and the deployment of novel classification methods into real world tools, I believe the field of adversarial defence is relevant.

- Authors use in a smart way the self-supervised loss which is traditionally used for learning good representation as pretrainining networks. I think self-supervised learning should extend its usage over the traditional framework and this paper is one example of its potential.

- Authors are also able to compute its own budget using the self-supervised loss, which I believe it's additional evidence that the usage of self-supervised learning for adversarial defence is interesting and useful.

- Quantitative results show how the proposed method is competitive with methods.

- Authors also evaluate the effectiveness of the method when faced with an attacker knowing which defence method is using. Although I consider the proposed attack a baseline attack (maybe other alternatives can be used), I believe it's a relevant result.


Weaknesses:
- I am missing evaluation of the method in larger scale datasets, or more natural images dataset. I think for this methods to be applicable and useful, authors should demonstrate its usefulness into real data.

- I am missing some images of the CIFAR dataset similar to Figure 2. I know the supplementary material shows some, but it would be good to include some into the main paper.

- I think authors should at least have one of the self-supervised methods (LC, RP or DR) show performance in both dataset. Given a new dataset, which methods should I select?

Conclusion: I believe the paper presents an interesting method with strong experimental results. The paper deserves acceptance as I believe it contains enough evidence to proof the effectiveness of the method.

---

> ### Author Response · Authors · 2020-11-23
> **Official Reply to Reviewer 3**
>
> We thank you for your positive feedback on our paper. We would like to address your remaining concerns accordingly:
>
> 1. We have included results for CIFAR100 in Table 3, demonstrating that we out-perform adversarial training on both l_2 and l_inf attacks and for two architectures (ResNet and WideResNet).
> 2. We have added a visualization of adversarial and purified images for CIFAR10. Note that in figure 2, we leverage the autoencoder in the DR auxiliary task to visualize how the network ‘sees’ the adversarial examples and purified images. There is no equivalent possibility for the RP and LC auxiliary tasks. Therefore we include the adversarial images and their purified versions.
> 3. We have added experimental results for CIFAR100, where LC mostly performs better than RP. An explanation for this is that LC is a more difficult task, and therefore leads to a “richer” representation space.
> In addition, note that there are practical reasons why some self-supervised tasks cannot work on both datasets. For example, RP will not work on MNIST because digits such as  0, 1, 6, 8 and 9 can be invariant/interchangeable to 0 and 180 degrees of rotation. We have included a discussion on this in the paper.
> We also want to stress that the scope of self-supervised tasks is not limited to the three we discussed. We encourage readers to explore more possibilities and choose one that is the most appropriate for the dataset.

---

> > ### Comment · AnonReviewer3 · 2020-11-24
> > **Thanks!**
> >
> > Dear Authors.
> >
> > Thank you very much for your reply, and for addressing my review comments.
> >
> > I think in my side all is clear now and I will keep my score.
> >
> > Thanks!

---

### Official Review · AnonReviewer4 · 2020-11-02
**reasonable and interesting idea, but needs more empirical validation**

**Rating:** 6
**Confidence:** 4

**Review:**

###################
Summary:

This paper studies adversarial defense by combing purification and self-supervised loss. During inference, the authors propose an online-purification method based on (clipped) iterative gradient ascent. The loss used by purification is from some pre-defined self-supervised tasks. During training, joint loss of softmax and self-supervised loss are used to match the purification process in inference. Experiments on MNIST10 and CIFAR10 demonstrate the effectiveness of the proposed method over several SOTA baselines. The evaluation considers both the white-box and black-box attack setup.

###################
Pros

1. The proposed method is well-motivated and reasonable.

2. The paper is clear and easy-to-follow.


###################
Cons

1. What is the T  for the online purification? Large T will significantly slow down the test time efficiency.

2. In Table 2, "FGSM AT" + "PGD", why it is "37.4%"? My understanding is this should be very small value, since multi-step PGD attack is pretty strong.

3. I am curious to see the gain by purely online-purification, maybe using the encoder by "PGD AT".

4. Seems like self-supervised tasks are pretty ad-hoc. Is there a principled way of selecting a good self-supervised task?

5. The two datasets used in the paper represents limited visual patterns. I think larger-dataset needs to be used, like cifar100, tiny imagenet.

######################### post-rebuttal

I appreciate the additional explanations and experiments by the authors. I also read the public discussion threads. I raise my score to 6. Two things for future:
- Make it work on bigger and more realistic images, imagenet, pascal, coco, etc. Now the adversarial community and deep learning community in general, highly relies on experiments, because theoretical guarantee is still mysterious. So we should push the field forward, by proving ideas on harder datasets.
- Explore stronger attacks, particularly gradient-free attack to avoid the obfuscated gradients.

---

> ### Author Response · Authors · 2020-11-23
> **Official Reply to Reviewer 4**
>
> We thank you for your valuable feedback on our paper. We would like to address your concerns accordingly
>
> 1. We agree that a large T will diminish test-time efficiency. As shown in the Sec. 4 Experiment, the value of T is 5 in each case which is relatively small. Practically, we found that increasing the value of T beyond 5 doesn’t help much. For more discussion on the choice of T, please see Sec. A2 in the appendix.
> 2. You are absolutely right that PGD is stronger than FGSM, but it is possible that FGSM-AT achieves relatively high PGD accuracy on CIFAR10 (e.g. SAT accuracy in table 1(c) from Song et.al. 2019). We believe the reason for this is that label leaking (Kurakin et.al. 2017), an issue standard FGSM-AT suffers from, does not happen in our case due to the low capacity of ResNet18. When label leaking happens, as in other cases in our paper, we see FGSM accuracy that is higher than No-Atk accuracy as well as low PGD accuracy (almost zero).
> 3. Our purification approach is purely online - the purification is performed at test-time. Also in contrast to Defense-GAN and Pixel-Defend which train the main model and the purifier separately, SOAP relies on the encoder f to learn a good auxiliary device h. It is not possible to use a different encoder with our auxiliary device h, since h operates with respect to the internal representation space (output of f), therefore the two need to be trained together. Thus a trained auxiliary device h cannot be simply composed onto a different network. We agree that it would be interesting to see how SOAP can be combined with AT for improved robustness in future work.
> 4. The whole field of self-supervised learning is growing fast but a theoretical foundation is still lacking, which makes such selection of appropriate self-supervised tasks empirical at this point. Nevertheless, we discuss some of the principles of selecting appropriate self-supervised tasks in Sec. 3.4. Our main suggestion is that the selection self-supervision should be differentiable (wrt inputs), comprehensive and efficient, and appropriate to the dataset. For example, RP will not work with MNIST since digits such as  0, 1, 6, 8 and 9 can be invariant/interchangeable to 0 and 180 degrees.
> 5. We have included experiments on CIFAR100 and we out-perform AT on both ResNet and WideResNet architectures, and on both l_2 and l_inf attacks.

---

### Public Comment · ~Nicholas_Carlini1 · 2020-11-10
**Defense idea built around obfuscating gradients**

This paper is explicit that "the reason [the defense is robust] is that the auxiliary component of the adapted attacks obfuscates the cross entropy gradient". It is well known (Athalye et al. 2018) that intentionally obfuscating gradient information is not a good technique to increase robustness. To the best of my knowledge, all papers that try this approach have been broken.

While the paper claims that "an adversary cannot benefit from the knowledge of the defense in a straightforward way", Tramer et al. 2020 has shown that in every case an adaptive attacker *can* take advantage of the particular details of a defense. It is necessary to actually demonstrate that adaptive attacks fail: claiming it without justification is insufficient.

---

> ### Comment · AnonReviewer2 · 2020-11-11
> **Thanks Nicholas, they have considered auxiliary-aware attacks**
>
> They have indeed considered adaptive attacks in the paper, see the section White-box attacks, paragraph "Auxiliary-aware attacks".
> It might be useful if you can share your thoughts on that, too (since you are already here).

---

> > ### Public Comment · ~Nicholas_Carlini1 · 2020-11-11
> > **Their consideration is saying adaptive attacks can't be done**
> >
> > It's under the section titled "Auxiliary-aware attacks" that the paper says "an adversary cannot benefit from the knowledge" and "the auxiliary component of the adapted attacks obfuscates the cross entropy gradient". So yes, while they have a section that tries to talk about it, the discussion is saying it can't be done.

---

> > > ### Comment · AnonReviewer2 · 2020-11-11
> > > **Thanks, my understanding is slightly different**
> > >
> > > I see your point. To me, for that paragraph, the authors indeed formulated a new objective that takes into consideration the knowledge of the auxiliary part, which is their equation (9). And they also mentioned in the beginning of the paragraph that,
> > > quote: *"In this section, we introduce a more ‘complete’ whitebox adversary which is aware of the purification method, and show that it is not straightforward to attack SOAP even with the knowledge of the auxiliary task used for purification."*
> > >
> > > From there, the authors found that there is still robustness w.r.t. this objective, so they conclude that, quote *"an adversary cannot benefit from the knowledge of the defense in a **straightforward** way"*.
> > >
> > > Of course, one can question whether this objective is sufficient for such a purpose, which is where I think you are heading at.

---

> > > > ### Public Comment · ~Nicholas_Carlini1 · 2020-11-12
> > > > **That may very well be true.**
> > > >
> > > > Agreed---It may very well be the case that an adversary can't benefit from this additional information. But if this turned out to be the case, it would be the first time I'm aware of this happening in this field. But every other defense that's made similar claims has shortly been broken by an adversary who has used this extra information.
> > > >
> > > > So the key question then is as you say: does the paper present a convincing argument that adversaries will fail at this task in the future, given knowledge of the defense?

---

> > > > > ### Comment · AnonReviewer2 · 2020-11-12
> > > > > **A difference of this work with previous "online" defenses**
> > > > >
> > > > > Agreed. Indeed that's the key question.
> > > > >
> > > > > Perhaps another useful and interesting point of this work is that the way it does "online defense", different from the previous work that I know of, is actually, again, **adversarial training**. This is encoded in the equations (3) and (4) in the draft. So at least intuitively, as long as ${\cal L}_{\text{aux}}$ is correlated with the main objective, this adversarial training works. And this is indeed why their equation (9) makes sense to me as a an adaptive attack (where they tried to explicitly break the correlation).
> > > > >
> > > > > On the flip side, this adversarial training also triggers one of my main questions in my review, quote: *"Following (1), what worries me more is that online-purification still needs to be aware of the attack type. Namely if one looks into equation (4), the objective has encoded norm-based attacks within it. This makes the results less interesting."*, and this actually leads to inferior adversarial robustness and increased inference cost (so the reduced training cost does not seem to be a good tradeoff).

---

> ### Author Response · Authors · 2020-11-17
> **Thank You and Some Clarification**
>
> We thank Nicholas for your attention and valuable comments on our paper, and we also want to thank Reviewer 2 for their clarification. (We will reply to all official reviewers in time along with the revised version of our paper. We did want to address Nicholas’s concerns during the public discussion period.)
>
> Reviewer 2 is correct about Section 4.1--we did not just claim SOAP is robust to auxiliary-aware / adaptive attacks but also proposed a targeted attack based on the full training model, including the auxiliary. This attack generates adversarial images from gradient ascent on the joint classification+auxiliary loss. The results are shown in Fig.4 for all three self-supervised auxiliary tasks.
>
> We also wanted to clarify that we did not design SOAP with the explicit purpose of ‘obfuscating gradients’, but this was a side comment with respect to the auxiliary-aware attack we presented, i.e., that it is hard for adversarial gradients to simultaneously increase cross entropy loss and maintain a small auxiliary loss.
>
> To reviewer 2, we would like to clarify (and will clarify this in the text of the submitted revision as well) that our defense is not based on knowledge of the attack. Rather we need to constrain the form of our defense (perturbation used to purify the image) in order to solve the optimization problem, so that the purified image resembles the natural input image, i.e. the perturbation is not unbounded. Alternative formulations to constrain the purification of Eq. 4 are beyond the scope of this work but an interesting topic for future work. Therefore, while our defense does parallel the l_inf bounded attack in its formulation, note that in our experiments we demonstrate it performs well on both l_inf bounded attack (FGSM, PGD) and l_2 bounded attack (CW, DF). (We will clarify and emphasize in the text that CW and DF reported in our results are l_2 bounded attacks). Therefore, although our purification is constrained based on an l_inf norm in Eq. 4, we show that SOAP can defend at the very least against l_inf and l_2 attacks. You could find some discussion on this topic in Sec. 3.3.

---

### Author Response · Authors · 2020-11-23
**General Reply**

We appreciate the detailed comments and recommendations provided by all reviewers. We have uploaded a new version of the paper that addresses issues brought up by all reviewers. First, we have added a new table presenting the results of applying SOAP to CIFAR-100. Notably, SOAP still achieves superior performance, regardless of the capacity of the network. In addition to Table 3, we have also provided further clarification in section 4.1 regarding the motivation for evaluating SOAP on white-box attacks, and additional interpretation of these results. We have also made clarifications regarding the choice of auxiliary tasks and scalability of the purification procedure, addressing points raised by Reviewer 2 and Reviewer 4.

---

### Decision · Program_Chairs · 2021-01-07
**Final Decision**

**Decision:**

Accept (Poster)

**Comment:**

This paper presents a defense scheme for adversarial attacks, called self-supervised online adversarial purification (SOAP), by purifying the adversarial examples at test time. The novelty of this work is in its incorporation of self-supervised representation learning into adversarial defense through purification via optimizing an auxiliary self-supervised loss. This is done by jointly training the model on a self-supervised task while it is learning to perform the target classification task in a multi-task learning setting. Compared with existing adversarial defense schemes such as adversarial training and purification techniques, SOAP has a lower computation overhead during the training stage.

**Strengths:**
  * It is novel to incorporate self-supervised learning for adversarial purification at test time.
  * SOAP’s training stage based on multi-task learning incurs low computation overhead compared with the original classification task.

**Weaknesses:**
  * Although the proposed adversarial defense scheme is computationally cheaper than the other existing methods during the training stage, it does incur some overhead during test time. This may be undesirable for some applications in which efficiency during test time is an important factor to consider.
  * The choice of a suitable self-supervised auxiliary task is somewhat ad hoc. The performance varies a lot for different auxiliary tasks.
  * The experimental evaluation is only based on relatively small and unrealistic datasets even after new experiments on CIFAR-100 have been added by the authors.

It is said in the paper that SOAP can exploit a wider range of self-supervised signals for purification and hence conceptually can be applied to any format of data and not just images, given an appropriate self-supervised task. However, this claim has not been substantiated in the paper using non-image data.

Despite some limitations and that some claims still need to be better substantiated, the paper presents some novel ideas which are expected to arouse interest for follow-up work in the adversarial attack and defense research community.